# Cumulative UV Exposure or a Modified SCINEXA™-Skin Aging Score Do Not Play a Substantial Role in Predicting the Risk of Developing Keratinocyte Cancers after Solid Organ Transplantation—A Case Control Study

**DOI:** 10.3390/cancers15030864

**Published:** 2023-01-30

**Authors:** Liliane Borik-Heil, Georg Endler, Walther Parson, Andreas Zuckermann, Lisa Schnaller, Keziban Uyanik-Ünal, Peter Jaksch, Georg Böhmig, Daniel Cejka, Katharina Staufer, Elisabeth Hielle-Wittmann, Susanne Rasoul-Rockenschaub, Peter Wolf, Raute Sunder-Plassmann, Alexandra Geusau

**Affiliations:** 1Department of Dermatology, Medical University of Vienna, 1090 Vienna, Austria; 2Department of Laboratory Medicine, Medical University of Vienna, 1090 Vienna, Austria; 3Institute of Legal Medicine, Medical University of Innsbruck, 6020 Innsbruck, Austria; 4Forensic Science Program, The Pennsylvania State University, University Park, PA 16801, USA; 5Department of Cardiac Surgery, Medical University of Vienna, 1090 Vienna, Austria; 6Department of Thoracic Surgery, Medical University of Vienna, 1090 Vienna, Austria; 7Department of Medicine III, Division of Nephrology and Dialysis, Medical University of Vienna, 1090 Vienna, Austria; 8Department of Nephrology, Ordensklinikum Barmherzige Schwestern Linz, 4020 Linz, Austria; 9Department of General Surgery, Division of Transplantation, Medical University of Vienna, 1090 Vienna, Austria; 10Department of Dermatology, Medical University of Graz, 8036 Graz, Austria

**Keywords:** keratinocyte cancer, solid organ transplant recipients, immunosuppression, UV burden, skin aging, *MC1R*

## Abstract

**Simple Summary:**

Due to life-long immunosuppression, organ transplant recipients are prone to skin cancer development. We evaluated the association of cumulative UV burden with skin aging and skin cancer in two groups of transplant patients with and without skin cancer, who were matched for gender, age, type of organ transplanted, post-transplantation period, and immunosuppressive therapy. Individuals with actinic keratoses had a 7.5-fold risk for skin cancer development, and those with green or blue eyes a 4.1- or 3.6-fold risk, respectively; carriers of particular *MC1R* genotypes associated with a diminished function had a 1.9-fold increased risk. The extent of skin aging was only connected with a higher number of tumors, but not with the development of skin cancer per se.

**Abstract:**

The risk of keratinocyte cancer is determined by intrinsic and extrinsic factors, which also influence skin aging. Few studies have linked skin aging and UV exposure with the incidence of non-melanoma skin cancer (NMSC). We evaluated signs of actinic skin damage and aging, individual UV burden, and melanocortin-1 receptor (*MC1R*) variants. A total of 194 organ transplant recipients (OTR) who suffered from NMSC were compared to 194 tumor-free controls matched for gender, age, type of transplanted organ, post-transplantation (TX) period, and immunosuppressive therapy. Compared with the cases, the controls scored higher in all skin aging scores and there were no differences in UV burden except for intentional whole-body UV exposure for specific UV scenarios and periods of life in favor of cases. The number of NMSCs correlated with all types of skin aging scores, the extent of intentional sun exposure, older age, longer post-TX period, shorter interval from TX to first NMSC, and specific *MC1R* risk groups. Multivariable models revealed a 7.5-fold risk of developing NMSC in individuals with actinic keratosis; 4.1- or 3.6-fold in those with green or blue eyes, respectively; and a 1.9-fold increased risk in the *MC1R* medium- + high-risk group. In the absence of skin aging contributing to NMSC development, certain *MC1R* risk types may identify OTR at risk for high tumor burden.

## 1. Introduction

Due to long-term immunosuppression (IS), organ transplant recipients (OTR), particularly of European descent, are at increased risk for skin cancer, especially non-melanoma skin cancer (NMSC) [1]. Cutaneous squamous cell carcinoma (cSCC) and basal cell carcinoma (BCC) are the most common malignancies after transplantation (TX) and account for more than 90% of all skin cancers in these patients [2,3,4]. Because the development of skin cancer is accelerated in OTR, they represent an ideal ’model’ for accelerated tumor development and skin cancer studies [5,6]. The pathogenesis of keratinocyte cancer (KC) is complex and involves multiple mechanisms [7]. Ultraviolet radiation (UVR) is thought to be the main driving force causing oxidative stress, IS, an inflammatory response, and DNA damage [8,9,10,11,12,13]. Clinically visible signs of actinic skin damage are an important factor reflecting past sun exposure. The SCINEXA™score is a validated visual score that has been shown in cohort studies to be well suited to measure the intrinsic and extrinsic aging of facial skin [14,15]. The extrinsic skin aging score is considered an indicator of the age-related cumulative effect of environmental factors on human skin, particularly UVR [15], and is synonymously referred to as ‘photoaging′, which is reflected in the structural and morphological changes of the skin. The causes of intrinsic or so-called chronological skin aging are less clear and correlate mainly with age, but they are also influenced by genetic factors. Intrinsically aged skin appears to be thinner, more lax, and more finely lined [16].

One of the key proteins involved in regulating skin pigmentation and thus protecting the skin from UV damage is the melanocortin-1 receptor (MC1R). Mutations in the *MC1R* gene result in decreased eumelanin production, leading to red hair color and increased susceptibility to NMSC due to impaired UV protection, which may be of greater importance in OTR under IS therapy [17,18,19]. 

In our study, we aimed to investigate an association between environmental and genetic factors and skin cancer incidence in a carefully selected cohort of OTR. We assessed individual UV burden, quantified skin aging using a validated photoaging score, and determined the *MC1R* genotype of the participants.

## 2. Patients and Methods

### 2.1. Patients

We enrolled 388 OTR (cases/controls, *n* = 194 each), recruited between 2017 and 2019. Cases who had suffered from at least one histologically proven NMSC (cSCC, BCC, Bowen′s disease (BD), Bowen′s carcinoma (BowCa)) post-TX were matched with 194 KC-free controls who had the same gender, age, were in the same post-TX period, and received comparable IS therapy. More male (149 pairs, 77%) than female subjects (45 pairs, 23%) signed up for this prospective case control study, with a mean age of 68 years at first visit (SD ± 9.1). The mean age at TX was 54.7 years (SD ± 12.2), and on average, participants had been transplanted 13.5 years before enrolment into the study (SD ± 7.8). The majority of OTR received a heart (HTR; *n* = 164; 42%) or kidney transplant (KTR; *n* = 148; 38%). Liver transplant recipients (LTR; *n* = 38; 10%) and lung transplant recipients (LuTR; *n* = 38; 10%) together accounted for 20%. The mean time from TX to the occurrence of the first NMSC was 7.9 years (SD ± 7.1) (Appendix A). A subset of 87 cases, again paired with 87 controls, had suffered 5 or more NMSC after TX.

We documented hair and eye color and classified skin photo type clinically into four categories—I, II, III, and IV—according to the Fitzpatrick skin type (FST: type I always burns, never tans; type II usually burns, tans minimally; type III sometimes burns slightly, tans evenly; type IV burns minimally, always tans well) [20,21]; skin types V and VI were not represented in the study cohort (Appendix A). Skin phototype was verified by a non-invasive method using a colorimeter. The melanin content of the skin at two sun-exposed sites (forehead and dorsum of the hands) and UV-unirradiated skin of the buttocks were determined according to the manufacturer′s recommendations, using reflectance spectroscopy to obtain a numerical and thus, quantifiable, value for melanin (the ‘melanin Index′ = MI) (DSM II Colorimeter™, Cortex Technology, Hadsund, Denmark; Appendix A) [22,23].

We quantified skin aging using a modified version of the SCore of INtrinsic and EXtrinsic skin aging (SCINEXA™score, intrinsic (ISA), extrinsic (ESA) both combined in the total skin aging score (TSA), Appendix A) [14,15,24] and assessed smoking history (Appendix A). This SCINEXA™score is a validated visual score for measuring intrinsic and extrinsic facial skin aging in case control studies [15]. Five signs characteristic of facial intrinsic skin aging (lax appearance, uneven pigmentation, fine wrinkles, decreased adipose tissue, and benign skin tumors, e.g., seborrheic warts or dermal nevi) and fifteen signs of extrinsic skin aging at defined locations were assessed and graded using ordinal scales as follows: 0 (none), 1 (mild), 2 (moderate), 3 (severe). This grading was applied to sunburn freckles (shoulders), solar lentigines (average score of 4 locations, i.e., neck, face, dorsum of hands, and forearms), pigment change, change in photo type of the skin, yellowness, pseudo-scars, coarse wrinkles (average value of the sites forehead, periorbital area, perioral area, nasolabial area), elastosis cutis, dryness, comedones, telangiectasias, and permanent erythema (neck/décolleté); for cutis rhomboidalis nuchae, Favre Racouchot′s disease, actinic keratosis, and the intrinsic item ‘uneven pigmentation′, a binary scale ‘present = 3’ and ‘absent = 0’ was used. The score was modified to exclude the variable ‘NMSC present′ because, per definition, only ‘cases′ but not ‘controls′ had tumors. The maximal score achievable for the ESA score was 45 and for the ISA score was 15 (Appendix A). In addition, we re-assessed all signs of intrinsic and extrinsic skin damage using photographs taken from each participant. Two dermatologists (AG, LBH) then scored these photographs independently of each other.

The individual UV burden of each patient was assessed using a standardized study protocol that focused on variables shown in Table 1 and Appendix A. Included were the place of residence during most of life taking into account altitude (<500 m, 500–1000 m, >1000 m; latitude was the same, as all study participants lived in Austria, and was therefore not included in further calculations) and ‘living >2 years outside Central Europe’ (north/south of Central Europe; in the subtropics/tropics).

Estimates of cumulative lifetime UV exposure are typically based on questionnaires of time spent outdoors, e.g., hours generally spent outdoors in sunlight during summer (May–August), i.e., between 9:00 AM and 3:00 PM, an approach that was also used in our study. To obtain reliable data on this topic, we carefully recorded the time spent outdoors on a typical weekday or weekend day. We additionally collected information on time spent outdoors for a specific activity (Appendix A). It is known that this allows individuals to provide more precise information, leading to an improved estimate of their actual sun exposure, as shown in a comparative study [25].

In addition, outdoor occupation was assessed (yes, no; if yes—hours spent with outdoor occupation); time spent outdoors during the week in leisure time and at work (May–August, Monday–Friday (hours) per decade/life stage: ages 10–19, 20–39, 40–59, ≥60 years were referred to as ‘UV scenario I’; time spent outdoors on weekends during leisure (May–August, (hours) per decade/life stage: ages 10–19, 20–39, 40–59, ≥60 were referred to as ‘UV scenario II′; UV scenario I + II together represent ‘total time spent outdoors with unintentional sun exposure’ (leisure plus occupation). ‘UV scenario III’ (sun exposure in Central Europe–sunbathing between 9:00 AM and 3:00 PM; hours per decade/life stage: ages 10–19, 20–39, 40–59, ≥60 years) and ‘UV scenario IV′ (sun exposure in southern geographic regions–vacation with high sun exposure in Mediterranean or subtropical/tropical regions, e.g., beach holiday; between 9:00 AM and 3:00 PM; hours per decade/life-stage: ages 10–19, 20–39, 40–59, ≥60 years) were combined into ‘UV scenario III + IV’: total time spent outdoors with intentional sun exposure (Table 1). 

Recreational activities independent of vacation included gardening and sunbed use–one unit sunbed use was calculated as 30 min (yes, no; if yes—total lifetime hours); recreational activities with body covered: mountaineering (>2000 m altitude), hiking, skiing/snowboarding (yes, no; if yes—total weeks of life); recreational activities wearing swimwear only: watersports, sunbathing (yes, no; if yes—total weeks of life); and recreational activities with uncovered body: nudist beach (yes, no; if yes—total weeks of life) (Table 2). The number of sunburns with skin peeling or blistering was assessed by frequency per decade (1–5, 6–10, 11–20, >20 times; ages 10–19, 20–39, 40–59, ≥60 years; Appendix A).

### 2.2. Mitochondrial DNA Point Heteroplasmy

In a subset of 23 individuals, skin scales from the sun-exposed preauricular area and from non-UV-exposed buttocks were prepared for the evaluation of UVR-induced mitochondrial DNA (mtDNA) point heteroplasmy (PHP). Only superficial layers adjacent to the stratum corneum and lacking vasculature were used, as it was essential to avoid contamination with blood for the analysis of PHP in keratinocytes only. Details of the mtDNA isolation and analysis of PHP can be found in the Appendix A. 

### 2.3. Genetic Analyses

Genomic DNA was isolated from the patients′ peripheral blood using the QIAsymphony^®^ (Qiagen, Hilden, Germany) according to the manufacturer′s suggestions. PCR and subsequent sequencing of the *MC1R* gene were performed using the following primers: for PCR, E1fw GCAGCACCATGAACTAAGCA, E1rv GTGGATGAAGCTTTCTGGTCA; for sequencing additionally, E1rva CTGCAGGTGATCACGTCAAT, E1rvb CAGGGTCACACAGGAACCA, and an Applied Biosystems 3130xl Genetic Analyzer (TFS) according to the manufacturer′s suggestions. 

Participants who did not carry an *MC1R* variant and individuals who carried variants that were unlikely to affect receptor function (based on published data or as predicted by bioinformatic mutation analysis tools) were considered as the *MC1R* 0/0 risk group. Together with heterozygous carriers of *MC1R* r variants (0/r; reduced receptor function), these individuals were assigned to the *MC1R* low-risk group (0/0 plus 0/r). A medium genetic risk for *MC1R* dysfunction was defined by the presence of either heterozygous R (0/R; loss of function), compound heterozygous Rr, or homo- or compound heterozygous rr variants. Individuals assigned a high *MC1R* risk carried homozygous or compound heterozygous RR variants [17,26].

### 2.4. Statistical Analyses

A statistical analysis was performed with SPSS 20 statistical software package (IBM SPSS Analytics, New York, NY, USA). Categorical variables were given as numbers and percentages continuous variables as the means and standard deviation unless otherwise stated. A comparison between groups was performed with a Chi-square test or a univariate ANOVA where appropriate. Correlation coefficients were calculated with a non-parametric Spearman′s correlation coefficient. A two-sided *p* value < 0.05 was considered as statistically significant. Due to the exploratory nature of our case-control design, we did not correct for multiple comparison. After the Bonferroni correction, the level of significance would be *p* < 0.01.

For a multivariate analysis to assess the major risk factors, a multivariate binary logistic regression model was applied including all major covariates (type of tumor, Fitzpatric skin type, hair color, eye color, *MC1R* genotype, freckles, moles, presence of actinic keratosis, smoking status, type of organ transplanted, skin aging score, and all major environmental variables (sun exposure, etc.)) in a reverse conditional model. Variables not reaching statistical significance at a *p* value > 0.1 were excluded. Odds ratios and the respective 95% confidence intervals are presented in a forest plot.

## 3. Results

### 3.1. Participants’ Characteristics

The characteristics of the participants (*n* = 388; cases/controls each *n* = 194) in terms of gender and age distribution, type of transplanted organ, age at TX, post-TX period, time to first NMSC after TX, IS regimen, and skin phenotypes including colorimetric assessment, and smoking history are shown in Appendix A. The mean value of the MI corresponded to each skin type, with higher values in darker skin (Appendix A). Regarding the pigmentation phenotype, of 388 participants, 9 (2%) were classified as FST I, the majority had FST II (*n* = 177, 46%) or FST III (*n* = 199, 51%); only 3 individuals had FST IV (1%). The distribution of FST differed significantly between cases and controls: more cases (55%) than controls (36%) had FST II, while controls had a higher proportion of type III and IV than cases (43% vs. 60%; *p* < 0.001; Appendix A).

The frequency and extent of sunburns were equally distributed between the cases and controls (Appendix A). Regarding smoking habits, there was no difference between cases/controls, neither in the whole study population (*p =* 0.056) nor in a subgroup of 97 FST-matched pairs (FSTII *n* = 43 pairs, FSTIII *n* = 54 pairs, *p =* 0.06; Appendix A).

### 3.2. Non-Melanoma Skin Cancer

Among the cases, a total of 1649 skin tumors were documented (1633 NMSC; 13 melanomas, 3 pleomorphic dermal sarcomas). Of the NMSC, 45% were BCC, 38% cSCC/BowCa, 17% were in situ carcinomas (BD); localizations and numbers of the different tumors per patient are shown in Figure 1, Appendix A. There was no correlation between the number of tumors and lighter skin and hair, smoking, sunburn freckles or moles, gender, or age at TX; however, there was a correlation with lighter eye color (*p =* 0.009), longer post-TX period (*p =* 0.01), and a shorter interval from TX to first NMSC (*p =* 0.019; (Appendix A).

### 3.3. Skin Aging Score

Cases had significantly lower ISA and TSA scores compared to the controls (*p =* 0.001 and *p =* 0.04, respectively; Figure 2A,B, Appendix A). The presence of AKs, which apart from solar lentigines was the only ESA score feature observed more frequently in cases (*p* < 0.001), did not shift the ESA score into significance (*p =* 0.236; Appendix A). However, all types of aging scores correlated significantly with the number of NMSC. Ninety-seven case–control pairs matched for FST (FST II *n* = 43 pairs, FST III *n* = 54 pairs), and thus having the same skin type was comparable with respect to age at examination, type of transplant, post-TX period, and smoking history (Appendix A). In these 97 pairs, all types of skin aging scores reached significance in favor of the controls (ISA score *p =* 0.034, ESA score *p =* 0.018, TSA score *p =* 0.016; Appendix A), as was also the case in a subset of 51 FST-matched and AK-free pairs. When comparing skin aging scores between individuals with FSTII and FSTIII in the 97 FST-matched pairs, only the ESA score was significantly higher in individuals with FSTII (*p =* 0.036; Appendix A), an effect not seen in the entire study population.

Higher values of all skin aging scores correlated with age >60 years (*p* < 0.001) and smoking (*p* < 0.005); male gender only correlated with the TSA score (*p =* 0.043). Cases/controls with smoking history did not differ in any of the aging scores, regardless of their FST (Figure 2C–E, Appendix A).

### 3.4. Mitochondrial DNA Point Heteroplasmy

In a pilot experiment, we obtained skin scales from the UV-exposed and -unexposed skin areas of 23 individuals. We found a statistically significant difference in the number of mutations in mitochondrial DNA between UV-unexposed and -exposed skin (*p* < 0.001; Appendix A). No correlation was observed between the extent of PHP and the level of the modified skin aging score and the extent of cumulative UV exposure.

### 3.5. Ultraviolet Burden

Outdoor occupational activities were not associated with the development of NMSC: 37% of cases and 44% of controls reported an outdoor occupation in the past (*p =* 0.180). We found no difference in skin aging scores among cases/controls with comparable occupational sun exposure. Control subjects spent more time outdoors with unintentional sun exposure (UV scenario I + II, i.e., body partly covered, with UVR exposure only to uncovered body parts such as head/neck, face, or upper extremities; *p =* 0.665; Table 1). Pairs additionally matched for TSA score (*n* = 57) showed no difference with respect to UV scenarios (Appendix A).

The total number of hours spent with intentional recreational and whole-body UV exposure (UV scenario III + IV) were significantly higher in cases (*p =* 0.033), due to increased UV exposure in different periods of life, e.g., sunbathing in Central Europe at ages 10–19 years (‘at home′, *p =* 0.027) and beach holidays in Southern/Mediterranean regions at ages 40–59 years (*p =* 0.017; Table 1).

The effect was more pronounced in 97 FST-matched pairs (UV scenario IV: *p =* 0.002; UV scenario III + IV: *p =* 0.015), and it remained significant in FST-matched pairs without AKs (*n* = 51 pairs; *p =* 0.023 and *p =* 0.037, respectively). When comparing 86 individuals with FSTII and 108 with FSTIII within the subgroup of 97 FST-matched pairs, we did not detect any difference in terms of UV exposure (Appendix A). The detailed survey of the weeks of life recreational activities showed no substantial differences between cases and controls (Appendix A).

In the whole study population, regardless of the development of NMSC, higher TSA scores correlated with higher unintentional (*p* < 0.001) as well as intentional (*p =* 0.007) sun exposure. Notably, unintentional sun exposure, including outdoor occupation, accounted for most of the total UV burden (Table 2). In the subgroup of 97 cases/control pairs additionally matched for FST, unintentional UV exposure was again associated with significantly higher TSA and ESA scores (*p =* 0.006 and *p =* 0.001, respectively; Appendix A). A higher ESA, but not TSA score, correlated with outdoor occupation (*p =* 0.040). For intentional sun exposure, only sunbathing in Central Europe correlated with an increased ESA score (*p =* 0.042). Mean lifetime hours of sunbed use and gardening were strongly correlated with TSA score (*p =* 0.014, *p =* 0.001, respectively; Appendix A). The total number of lifetime hours of UV exposure was twice as high in Central Europe as in the southern regions, explaining the association with higher skin aging scores (ESA and TSA *p =* 0.001, ISA score *p =* 0.036) (Table 1 and Table 2). Smoking had an additive effect on UV-related skin aging: in the subgroup of cases with a UV burden in the highest quartile (*n* = 97), we found a statistically significant higher ISA and TSA score in smokers compared to non-smokers (*n* = 62; ISA score: *p =* 0.003, TSA score *p =* 0.015; Appendix A).

### 3.6. Analyses of Cofactors for Non-Melanoma Skin Cancer

In the cases, the total numbers of NMSC (all locations), cSCC (total, head/neck, and trunk region), and BCC (total, head/neck region, upper extremities) correlated with all types of skin aging scores (range *p* < 0.001 to *p =* 0.025), although we evaluated skin aging in the head/neck region only (Appendix A). When we linked the number and types of NMSC with UV exposure, occupational sun exposure correlated only with the total number of cSCC (*p =* 0.012), especially in the head/neck region (*p =* 0.001); no significant correlations were observed for BCC in this setting (Figure 3, Appendix A).

The total time spent outdoors with unintentional sun exposure (UV scenario I + II) showed no correlation with the total number of NMSC, but it was highly and exclusively correlated with the total number of cSCC (*p =* 0.001) and the number of SCC in the head/neck location (*p* < 0.001). The total time spent outdoors with intentional sun exposure (uncovered body, wearing swimwear, UV scenario III + IV) was highly correlated with the total number of NMSC (*p =* 0.002), cSCC (*p* = 0.049), and BD (*p* = 0.025); with respect to the different tumor locations, only the trunk region reached significance (*p* = 0.009) for cSCC numbers, and for BCC, the lower extremities (*p* = 0.025; Figure 3, Appendix A).

Outdoor recreational activities or gardening showed no correlation with cSCC or BCC, while mountaineering, hiking, or skiing correlated with the occurrence of BD in the head/neck area (*p* = 0.031), and sunbed use correlated with BD in the lower extremities (*p* = 0.045). When comparing the high-risk cases (≥5 NMSC) of FSTII (*n* = 48) with those of FSTIII (*n* = 35), no difference was observed regarding UV burden in the different UV scenarios. When comparing these two groups in terms of skin aging, only the ESA score was significantly higher in cases with FSTII compared to FSTIII (*p =* 0.042; Appendix A).

In the subgroups of 87 cases with ≥5 NMSC and 44 cases with ≥10 NMSC—compared to those with a lower number of NMSC (1–4 NMSC, 1–9 NMSC, respectively)—higher values in all skin aging scores correlated with a higher number of NMSC, except for the ISA score in cases ≥10 NMSC (Figure 2F,G). Only intentional vacation-associated high UV exposure correlated with a higher number of NMSC (≥5 NMSC: *p =* 0.004; ≥10 NMSC: *p =* 0.002). The type of activity, full or partial sun exposure, or sunbed use made no difference (Table 3).

### 3.7. MC1R Analyses

The *MC1R* genotype was available for each patient included in this study (*n* = 388; Appendix A). The definition of *MC1R* risk variants/groups is described in the Appendix A. Red hair was found exclusively in R/R individuals. Individuals with FST I predominantly (89%) carried R/R high-risk variants (*n* = 8), whereas the majority of participants with FST II and III carried no (0/0; 30% and 38%, respectively) or low-risk 0/r (32% and 34%, respectively) variants (Appendix A).

The cases and controls differed significantly in their distribution within the low- and medium- + high-risk groups (*p =* 0.03). The only *MC1R* haplotype associated with NMSC was defined by the relatively rare variant c.464T > C (*p =* 0.03; Appendix A). Only the risk group 0/R was more common among cases (22%) compared to controls (14%; *p =* 0.044; Appendix A), and also more common in cases with FSTII (27% cases versus 15% controls).

Higher NMSC counts clearly correlated with the *MC1R* medium- + high-risk group when cases with a history of ≥5 or ≥10 NMSC were compared with those with 1–4 (*p =* 0.004) or 1–9 (*p =* 0.001) NMSC, respectively, or with matched controls (*p =* 0.008 and *p =* 0.019, respectively). The associations of NMSC counts, *MC1R* risk groups, and FST are shown in Figure 4. In contrast to the overall study population (*p =* 0.325), in the subset of cases with a high tumor burden (≥10 NMSC), individuals with FSTIII or dark brown hair more frequently carried medium + high-risk variants than cases with ≤10 tumors (*p =* 0.008 and *p =* 0.001, respectively). This suggests a pigmentation-independent effect of *MC1R* on the tumor number. Only in blue-eyed individuals, did cases carry medium- + high-risk variants more frequently (43%) than controls (25%; *p =* 0.024), an observation that was more pronounced when cases of ≥5 or ≥10 tumors were compared with matched controls (*p =* 0.007 and *p =* 0.028, respectively; Appendix A).

Regarding the numbers of different NMSC entities, only the number of cSCC was significantly and exclusively higher in the RR risk group (*p* = <0.001). However, due to the small number of patients, the informative value of this observation is limited.

Skin aging scores gradually increased from the low, medium, and high *MC1R* risk groups in the whole study population (ESA score *p =* 0.002; TSA score *p =* 0.01) and when calculated separately in cases and controls. Only the ESA score was significantly higher in cases in the high *MC1R* risk group (RR, *p =* 0.02; Figure 4), but again, there was only a small number of cases.

Within the different *MC1R* risk groups, self-reported UV exposure was comparable between the cases and controls (Appendix A). Therefore, an additional UV effect on the development of skin cancer in carriers of medium- and high-risk *MC1R* variants can be neglected.

A multivariate analysis revealed a 1.9-fold increased risk for subjects belonging to the medium- + high-risk *MC1R* group, and a 4.1- and 3.6-fold increased risk for individuals with green or blue eyes, respectively, while the presence of AKs conferred a 7.5-fold increased risk for the development of NMSC (Figure 5).

## 4. Discussion

Organ transplant recipients are at high risk for KC, particularly cSCC [2], which accounted for 55% of the 1633 NMSC in our study. The selection of two cohorts of patients —‘cases’ with NMSC and matched ‘controls′ who remained tumor-free after TX—provided us with an approach to evaluate the effect of skin aging and UV exposure on the development of NMSC in OTR. A valid assessment of lifetime UVR exposure and skin aging using a standardized questionnaire and a detailed survey of the signs of skin aging was a prerequisite for the evaluation of factors contributing to the development of NMSC.

Extrinsic skin aging, synonymously referred to as ‘photoaging’, i.e., structural and morphological changes of the skin including the clinical sign of AKs, serves as an indicator of age-related environmental effects on human skin, especially UVR [15]. Apart from solar lentigines, AKs were the only ESA score parameter that was significantly higher in cases: nearly 50% were affected, compared with 16% in the controls. All other ESA score parameters and the TSA score reached higher values in the controls. Concerning the whole study population, higher scores correlated with age, male gender, and history of smoking, independent of FST and/or the presence of AKs, demonstrating the reliability of our quantification of skin aging [27,28]. We observed a non-significant trend toward higher ESA score values in individuals with FSTII compared with FSTIII, with more than 50% of cases being FSTII compared with 36% of the controls: a lighter skin type seems to be associated with the development of NMCSs after TX. To avoid bias due to skin type, we examined cases/control pairs that were additionally matched on FST. Again, the controls scored higher on all skin aging scores, suggesting that skin aging per se does not necessarily predict a higher risk for KC. In addition to the exogenous factors, interindividual differences in skin aging can also be attributed to genetic factors that influence collagen degradation and elastin deposition [29,30]. The intrinsic skin aging score may not adequately reflect these factors, which could explain the lack of association between skin aging and KC development.

Only in the *MC1R* high-risk group (R/R) did cases have higher ESA and TSA scores. *MC1R* is an important determinant of skin phototype, as it regulates melanin production [26,31,32,33,34,35,36,37,38,39,40,41,42,43,44]. While FST appears to be the major factor predisposing to skin cancer in our collective, the *MC1R* genotype seems to predominate over FST: in a multivariate analysis, the *MC1R* genotype, but not the FST, remained significant. In our cohort and consistent with published data, FSTI correlated with ‘high-risk types’, i.e., R/R *MC1R* carriers [43]. Only in cases that had FSTII and/or blue eyes, did the medium- + high-risk *MC1R* group correlate with the development of NMSC, confirming a predisposition to KC in individuals with a lighter skin type. Nevertheless, a higher percentage of cases and the majority of controls carried low-risk variants, implying that additional genetic factors play a role. We found a correlation between *MC1R* risk types and the number of skin tumors in individuals with a history of more than 5 or 10 NMSCs. Those with a higher tumor burden more often carried medium- or high-risk *MC1R* variants. Interestingly, this was also true for cases with FSTIII and/or dark brown hair, indicating a pigmentation-independent effect of *MC1R* variants on KC development. Tagliabue et al. reported similar results [43].

While we assessed skin aging signs only in the head/neck area, skin aging scores correlated with higher numbers of NMSC at all locations, numbers of cSCC predominately in the head/neck/trunk area, and BCC in the head/neck/upper extremities region. This suggests that our quantification of skin aging provides a risk assessment for the occurrence of multiple NMSCs in the most commonly affected areas.

We applied established protocols for the reliable assessment of sun exposure and collected information on time spent outdoors and outdoor activities [25]. Comparing cases and controls, we found no difference in unintentional or occupational UV exposure—the latter possibly because of the small sample size—nor any association with different leisure activities. Sunbathing in Central Europe in adolescence, and beach vacations in the southern or Mediterranean regions in adulthood, appear to be associated with the development of NMSC and higher tumor count, independent of FST. This implies that high-intensity UVR is critical for the development of KC [45].

To date, there are no data on occupational sun exposure in OTR [46]. In our study, the number of cSCC—but not BCC—in the head/neck region correlated with occupational UV exposure. Studies focusing on NMSC in association with cumulative occupational UV exposure are conflicting [46,47]. They report an increased risk of NMSC in outdoor workers, mostly farmers, and differ for cSCC and BCC [46,48]. A meta-analysis found a stronger association with occupational UV exposure for BCC than for cSCC [46]. This is supported by the fact that in the general population the incidence of BCC is twice that of cSCC [49], whereas in OTR the ratio is inverse.

Higher intentional and unintentional UV exposure correlated with higher TSA scores [28]. In a subset of pairs matched for skin aging—30% of the cohort—we found an equal UV burden, demonstrating that UV exposure contributes to skin aging [12] without clear evidence of its effect on NMSC. A subanalysis of pairs additionally matched on FST confirmed this. The number of NMSCs correlated with aging scores and specific UV exposure. There was no correlation with lighter skin type or smoking history.

Our findings are consistent with the knowledge that UVR is a major factor in age-related skin changes that are accelerated by smoking [27]. In our study, all aging scores were higher in smokers, which remained significant in 97 FST-matched pairs. However, there was no significant difference between cases and controls in this subgroup of 97 FST-matched pairs with respect to smoking, suggesting that smoking plays a role in skin aging but does not appear to be associated with cancer development [27,50].

Consistent with the literature, AKs as precursors of cSCC were highly associated with UVR, but only in the context of intentional sun exposure in Central Europe [51]. Spending time outdoors, fully clothed but with the head and neck uncovered, correlated exclusively with high numbers of cSCC in this region. Intentional UV exposure correlated strongly with the total number of NMSC as well as cSCC on the trunk but not on the head/neck. In this UV scenario, we found a correlation between the total number of BD and BCC on the lower extremities. Outdoor recreational activities or gardening showed no particular association with NMSC.

To investigate an association between the extent of UV exposure and MC1R genotypes was of interest because there might be an additive effect on skin tumor development. However, we found no difference between our cases and the controls when we stratified MC1R genotypes and UV exposure with tumor development. A novel aspect was a possible association of *MC1R* risk types with the individual diversity of KC types; cases that developed a single tumor entity preferentially carried low-risk variants, whereas cases affected by the full spectrum of KC had equal distributions of low- and medium- + high-risk *MC1R* types. Finally, in multivariable models, we identified AKs, green or blue eyes, and the presence of medium- + high-risk *MC1R* types as factors associated with the development of NMSC. Skin aging, time spent outdoors, sunbed use, FST, and hair color had no significant association in these models.

To our knowledge, this is the only case control study in OTR with and without skin tumors in which skin aging, previous UV exposure, and *MC1R* genotypes were evaluated. The strength of this study lies in the fact that patients with and without skin tumors were maximally matched in patient characteristics, post-TX period, and immunosuppressive therapy. Some limitations should be considered: due to the case-control design, selection bias cannot be completely excluded, and because of the exploratory nature of the case-control design, our results should be confirmed in a cohort study.

## 5. Conclusions

In analyzing a substantial number of OTR who received IS therapy and were therefore at risk for developing a skin tumor, we found that in our study population, the actual risk for developing KC was not adequately explained by the extent of skin aging, UV-related skin damage, or cumulative UV exposure. However, skin aging does influence the individual number of tumors. Previous intensive UV exposure was relevant to the development of KC. Unintentional sun exposure accounted for most of the total UV burden: twice as high in Central Europe as in the southern regions, explaining the association with higher skin aging scores. Whole-body UV exposure was associated with a higher number of NMSCs. The same was observed for certain MC1R variants, even in individuals with FSTIII, suggesting a pigmentation-independent effect (Table 4).

To summarize, in OTR, skin aging increases with age, smoking history, and high UVR exposure. Men have higher skin aging scores. Risk factors for developing NMSC include lighter phenotype, holiday-related high UVR, specific *MC1R* variants, and the presence of AK, but not skin aging per se. The number of NMSCs increases with longer time after TX period, shorter interval between TX and first NMSC, older age, higher skin aging scores, the presence of AK, specific *MC1R* variants, and severe sunburns as young adults.

Independently of the known proactive measures in the dermatologic follow-up of OTRs, to better assess the risk for developing skin tumors, a history regarding previous high-intensity UVR exposure could be performed. The inclusion of genetic factors could be an additional facet of risk assessment in the future.

## Figures and Tables

**Figure 1 cancers-15-00864-f001:**
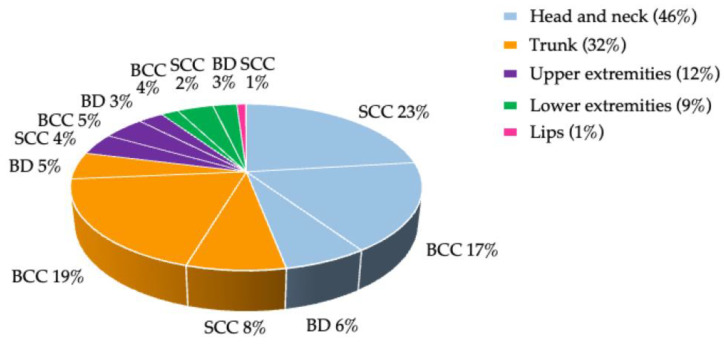
Anatomical localization of NMSC subtypes.

**Figure 2 cancers-15-00864-f002:**
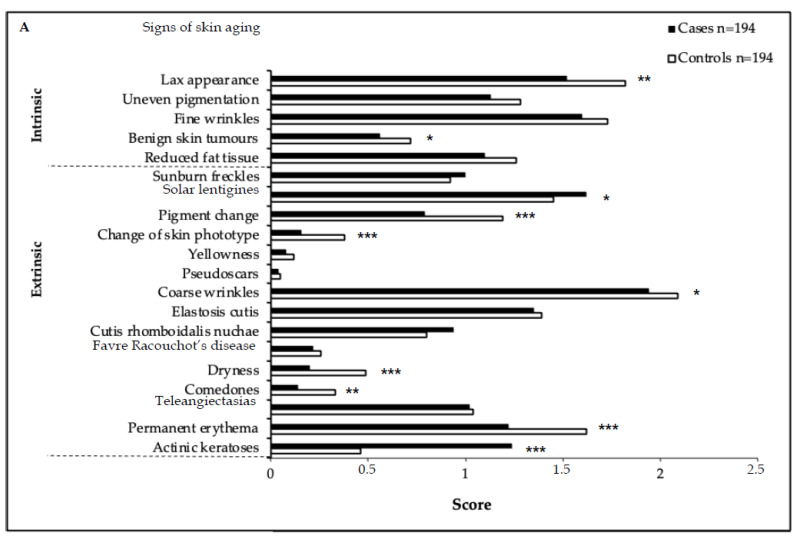
Signs of skin aging and skin aging scores. Significant *p* values are indicated by asterisks: * *p* < 0.05, ** *p* < 0.01, *** *p* < 0.001. (**A**) Signs of skin aging in cases and controls; (**B**) skin aging scores (mean ± standard deviation) in cases and controls; (**C**) in age groups ≥60 (*n* = 310) and <60 (*n* = 78) years at time of evaluation; (**D**) in males (*n* = 298) and females (*n* = 90); (**E**) correlated with smoking history: smokers (*n* = 243) and non-smokers (*n* = 145); (**F**) in cases with ≥5 NMSCs (*n* = 87) and cases with 1–4 NMSCs (*n* = 107); (**G**) in cases with ≥10 NMSCs (*n* = 44) and cases with 1–9 NMSCs (*n* = 150).

**Figure 3 cancers-15-00864-f003:**
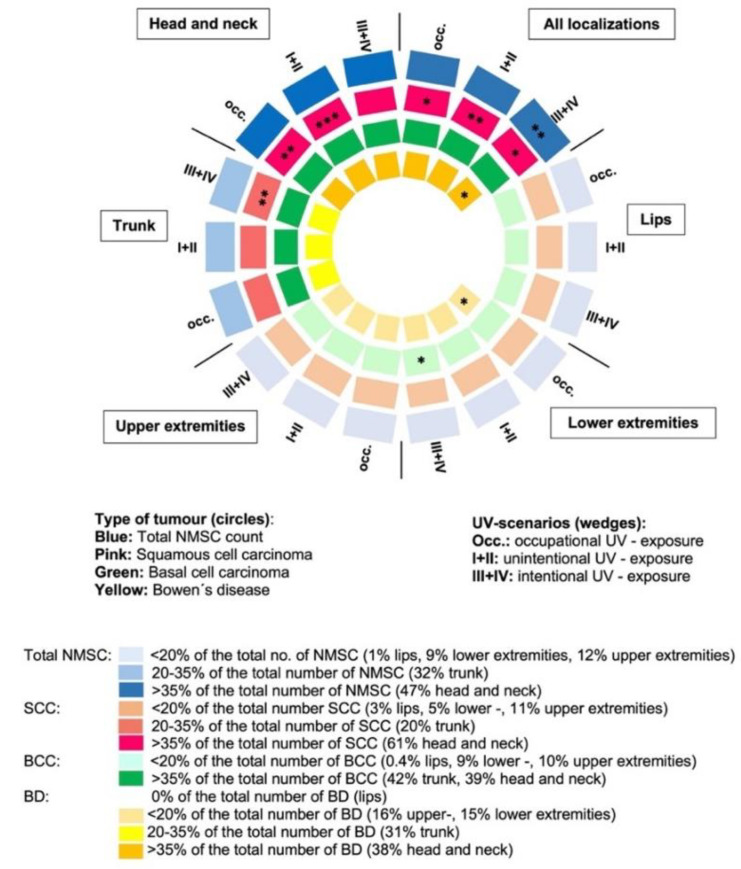
Correlations between UV exposure, localization, and number of NMSCs. NMSC types are shown as colored circles and the percentage of NMSCs as colored wedges, with the intensity of the colors corresponding to the percentage of the total number of each tumor. Significant *p* values are indicated by asterisks, *: *p* < 0.05, **: *p* < 0.01, ***: *p* < 0.001.

**Figure 4 cancers-15-00864-f004:**
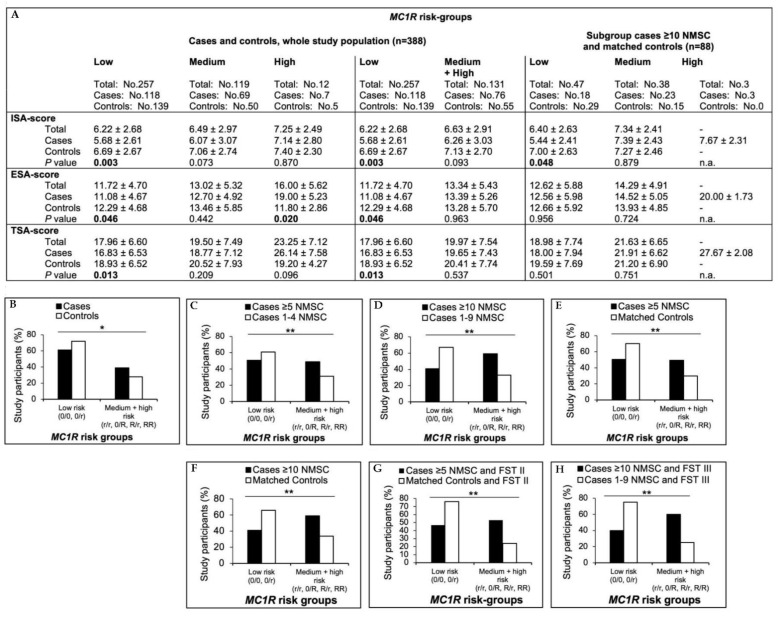
Distribution of *MC1R* risk groups in different cases/controls scenarios: (**A**) correlated with skin aging scores; significant *p* values (*p* < 0.05) calculated by Chi-square test are displayed in bold; n.a.: not available, ISA: intrinsic skin aging, ESA: extrinsic skin aging, TSA: total skin aging; (**B**) in the entire study population, *: *p* = 0.03; (**C**) in cases with ≥5 NMSC (*n* = 87) and cases with 1–4 NMSC (*n* = 107), **: *p* = 0.004; (**D**) in cases with ≥10 NMSC (*n* = 44) and cases with 1–9 NMSC (*n* = 150), **: *p* = 0.01; (**E**) in cases with ≥5 NMSC (*n* = 87) and matched controls (*n* = 87), **: *p* = 0.008; (**F**) in cases with ≥10 NMSC (*n* = 44) and matched controls (*n* = 44), **: *p* = 0.019; (**G**) in cases with ≥5 NMSC and FST II (*n* = 48) and matched controls and FST II (*n* = 34), **: *p* = 0.008; (**H**) in cases with ≥10 NMSC and FSTIII (*n* = 15) and cases with 1–9 NMSC and FSTIII (*n* = 68), **: *p* = 0.008.

**Figure 5 cancers-15-00864-f005:**
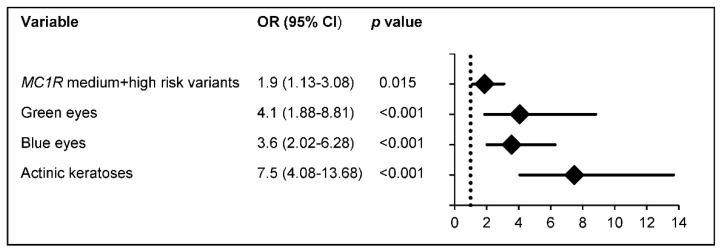
Multivariate logistic regression analysis for factors contributing to the risk of NMSC development. The x-axis represents the odds ratios (diamonds) and 95% confidence intervals (solid horizontal lines). The dashed vertical line indicates an OR value of 1 (no effect). Factors identified in multivariable models and significantly associated with NMSC development included blue eyes (OR: 3.6; 95% CI: 2.02–6.28; *p* < 0.001*), green eyes* (OR: 4.1; 95% CI: 1.88–8.81; *p* < 0.001), *MC1R* medium- + high-risk variants (OR: 1.9; 95% CI: 1.13–3.08; *p* = 0.015), and AKs (OR: 7.5; 95% CI: 4.08–13.68; *p* < 0.001).

**Table 1 cancers-15-00864-t001:** UV exposure in lifetime hours and UV scenarios in cases and controls.

	Total	Cases	Controls	*p* Value ^#^
	*n* = 388	*n* = 194	*n* = 194	
**Main place of residence** (altitude; *n* (%))				0.594
<500 m	370 (95)	187 (96)	183 (94)	
500–1000 m	16 (4)	6 (3)	10 (5)	
>1000 m	2 (1)	1 (1)	1 (1)	
**Residence abroad > 2 years** (*n* (%))				**0.041**
Northern Europe	7 (2)	4 (2)	3 (2)	
Southern Europe	41 (11)	15 (8)	26 (13)	
Subtropics	29 (8)	10 (5)	19 (10)	
Tropics	7 (2)	6 (3)	1 (1)	
**Time spent outdoors with unintentional sun exposure***(May*–*August)*				
**Outdoor occupation**				
All study participants (hours, *mean ± SD)*	12,257 ± 10,355	12,246 ± 10,217	12,268 ± 10,589	1.000
Participants with outdoor occupation (*n* (%))	157 (40)	72 (37)	85 (44)	0.180
Without outdoor occupation (*n* (%))	231 (60)	122 (63)	109 (56)	0.180
**Recreation and occupation during the week (UV scenario I**; hours, *mean ± SD)*				
	17,369 ± 12,868	17,080 ± 12,903	17,657 ± 12,836	0.659
Ages 10–19 years	3604 ± 1748	3513 ± 1735	3703 ± 1761	0.286
Ages 20–39 years	5699 ± 6259	5629 ± 7033	5769 ± 4939	0.820
Ages 40–59 years	5345 ± 7697	5285 ± 8497	5377 ± 6897	0.908
Ages ≥60 years	2993 ± 2462	2950 ± 2330	3110 ± 2593	0.550
**Recreation on weekends (UV scenario II;** hours*, mean ± SD)*				
	10,827 ± 3778	10,782 ± 3428	10,871 ± 4128	0.814
Ages 10–19 years	2475 ± 607	2463 ± 593	2487 ± 621	0.688
Ages 20–39 years	3962 ± 1522	3995 ± 1461	3929 ± 1582	0.670
Ages 40–59 years	3172 ± 1555	3132 ± 1494	3196 ± 1647	0.680
Ages ≥60 years	1337 ± 1109	1316 ± 1039	1390 ± 1178	0.540
**Total time spent outdoors with unintentional sun exposure (UV scenario I + II**; total hours *mean ± SD)*	28,196 ± 15,163	27,862 ± 14,571	28,528 ± 15,755	0.665
**Time spent outdoors with intentional sun exposure***(9:00 AM–3:00 PM,* *May*–*August; mean ± SD)*				
**Central Europe (UV scenario III;** *hours, mean ± SD)*				
	5678 ± 3946	6023 ± 3699	5333 ± 4192	0.087
Ages 10–19 years	1951 ± 1197	2087 ± 1180	1816 ± 1213	**0.027**
Ages 20–39 years	1875 ± 1443	1949 ± 1396	1801 ± 1489	0.313
Ages 40–59 years	1341 ± 1354	1454 ± 1371	1221 ± 1337	0.092
Ages ≥60 years	598 ± 1455	610 ± 1068	564 ± 1304	0.750
**Southern geographic regions (UV scenario I*V****, Mediterranean, subtropical, tropical regions*; hours, *mean ± SD)*				
	2774 ± 2610	2903 ± 2558	2586 ± 2622	0.228
Ages 10–19 years	288 ± 647	253 ± 543	323 ± 750	0.294
Ages 20–39 years	1148 ± 1096	1189 ± 1158	1107 ± 1034	0.464
Ages 40–59 years	995 ± 1054	1120 ± 1049	863 ± 1058	**0.017**
Ages ≥60 years	369 ± 718	385 ± 663	337 ± 772	0.590
**Total time spent outdoors with intentional sun exposure (UV scenario III + IV**; total hours*, mean ± SD)*	8423 ± 4645	9019 ± 4771	8228 ± 4519	**0.033**
**Recreational activities independent of vacation** (except gardening; hours, *mean ± SD)*	3398 ± 4053	3136 ± 3471	3592 ± 4635	0.275
**Gardening** (hours, *mean ± SD*)				
All study participants	3109 ± 4971	2992 ± 4362	3227 ± 5579	0.645
Gardeners only	4340 ± 3812	4206 ± 3250	4471 ± 4375	0.700
*n* (%)	278 (72)	138 (71)	140 (72)	
**Sunbed use** (hours, *mean ± SD)*				
All study participants	6 ± 23	7 ± 33	5 ± 12	0.131
Sunbed users only	34 ± 10	33 ± 12	36 ± 8	0.392
*n* (%)	66 (17)	40 (21)	27 (14)	

**^#^** Significant *p* values (*p* < 0.05) by two-sided Student’s *t*-test are displayed in bold. SD: standard deviation.

**Table 2 cancers-15-00864-t002:** Correlations of the skin aging scores with different UV scenarios in the whole study population.

		Skin Aging Score		
		Intrinsic	Extrinsic	Total
**Time spent outdoors with unintentional sun exposure** *(May*–*August; mean lifetime hours)*				
**Outdoor occupation**	** *r_s_* **	−0.021	0.100*	0.068
	***p* value**	0.678	**0.048**	0.182
**Total time spent outdoors with unintentional sun exposure** (Recreation and occupation; **UV scenario I + II**)	** *r_s_* **	0.129*	0.231 **	0.217 **
	***p* value**	**0.011**	**<0.001**	**<0.001**
**Time spent outdoors with intentional sun exposure** *(9AM–3PM, mean lifetime hours)*				
**Central Europe (UV scenario III)**	** *r_s_* **	0.107*	0.167 **	0.166 **
	***p* value**	**0.036**	**0.001**	**0.001**
**Southern geographic regions (UV scenario IV)**	** *r_s_* **	0.015	−0.031	−0.026
	***p* value**	0.765	0.538	0.611
**Total time spent outdoors with intentional sun exposure** **(UV scenario III + IV)**	** *r_s_* **	0.110*	0.132 **	0.137 **
	***p* value**	**0.030**	**0.009**	**0.007**
**Recreational activities during vacation** *(mean weeks of life)*				
**Body covered** (mountaineering, hiking, skiing)	** *r_s_* **	0.073	0.083	0.091
	***p* value**	0.153	0.101	0.073
**Wearing swimwear only** (watersports, sunbathing)	** *r_s_* **	0.004	0.074	0.056
	***p* value**	0.931	0.148	0.271
**Body uncovered** (nudist beach)	** *r_s_* **	0.041	0.046	0.042
	***p* value**	0.421	0.368	0.405
**Sunbed use** *(mean lifetime hours, sunbed users only)*	** *r_s_* **	0.318 **	0.323 **	0.362 **
	***p* value**	**0.009**	**0.008**	**0.003**
**Sunbed use** *(yes/no)*	** *r_s_* **	−0.163 **	−0.080	−0.130 *
	***p* value**	**0.001**	0.116	**0.010**
**Gardening** *(mean lifetime hours)*	** *r_s_* **	0.166 **	0.195**	0.213 **
	***p* value**	**0.005**	**0.001**	**<0.001**

*r_s_*: Spearman′s rank correlation coefficient. *: correlation. **: strong correlation. Significant *p* values (*p* < 0.05) calculated by Spearman’s rank correlation coefficient are displayed in bold.

**Table 3 cancers-15-00864-t003:** Total hours of UV exposure in different UV scenarios in subgroups of cases.

	Subgroups of Cases					
	1–4 NMSC	≥5 NMSC	*p* Value	1–9 NMSC	≥10 NMSC	*p* Value
	*n* = 107	*n* = 87		*n* = 150	*n* = 44	
**Time spent outdoors with unintentional sun exposure** (*May*–*August;* hours, *mean ± SD*)						
**Outdoor occupation**	4662 ± 8927	4402 ± 8178	0.834	4949 ± 9046	3167 ± 6652	0.227
**Total time spent outside with unintentional sun exposure (UV scenario I + II;** total hours, *mean ± SD*)	26,082 ± 10,998	30,051 ± 17,850	0.059	27,843 ± 14,225	27,927 ± 15,869	0.973
**Time spent outdoors with intentional sun exposure** *(May*–*August;* hours, *mean ± SD*)						
**Central Europe (UV scenario III)**	5277 ± 3261	6940 ± 4010	**0.002**	5693 ± 3534	7147 ± 4062	**0.022**
**Southern geographic regions** **(UV scenario IV** *; Mediterranean, subtropical, tropical regions)*	2762 ± 2305	3076 ± 2842	0.397	2658 ± 2287	3739 ± 3210	**0.013**
**Total time spent outside with intentional sun exposure** **(UV scenario III + IV;** total hours, *mean ± SD)*	8040 ± 4040	10,017 ± 5464	**0.004**	8351 ± 4296	10,886 ± 5938	**0.002**
**Gardening** (hours*, mean ± SD*)	4076 ± 4810	4386 ± 4467	0.701	3963 ± 4487	5161 ± 5240	0.225
**Holiday activities** (weeks, *mean ± SD*)						
**Body covered (mountaineering, hiking, skiing)**	69 ± 107	59 ± 73	0.472	65 ± 97	62 ± 78	0.808
**Wearing swimwear (watersports, sunbathing)**	44 ± 76	39 ± 61	0.608	42 ± 71	41 ± 67	0.899
**Full body uncovered (nudist beach;** weeks, *mean ± SD*)	3 ± 13	8 ± 31	0.131	5 ± 24	6 ± 21	0.830

Significant *p* values (*p* < 0.05) by two-sided Student’s *t*-test are displayed in bold. SD: standard deviation.

**Table 4 cancers-15-00864-t004:** Factors associated with skin aging, NMSC development, and NMSC number.

	Skin Aging Whole Study Population	NMSC Development Cases Versus Controls	NMSC Numbers Cases
**Demographic data**	>60 years Smoking history Male gender		Longer post-TX period Shorter interval from TX to first NMSC Older age at examination
**Skin type, hair, and eye color**		FST II, lighter hair, and blue and green eyes ^#^	Lighter eye color
**Skin aging**		^+^	High aging scores
**UVR**	Intentional and unintentional high UVR exposure	Vacation-related high intensity UVR *	Sunburns with blistering at age 20–39 Outdoor occupation: cSCC in the head/neck region BD on the lower extremities
** *MC1R* **		Specific MC1R variants	Specific *MC1R* variants
**Actinic keratosis**		AK	AK

^#^ in FST-matched cases and controls, no difference; ^+^ ISA and TSA higher in controls, ESA comparable; * no difference in the cumulative UVR between cases and controls.

## Data Availability

The data that support the findings of the study are available upon request.

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
