# Peer review of "Cumulative UV Exposure or a Modified SCINEXA™-Skin Aging Score Do Not Play a Substantial Role in Predicting the Risk of Developing Keratinocyte Cancers after Solid Organ Transplantation—A Case Control Study"

_cancers, 2023, doi:10.3390/cancers15030864_

Round 1
Reviewer 1 Report
In order to identify risk factors for NMSC additional to immunosuppression in OTR the authors have evaluated the association of skin aging and UV exposure with the incidence of keratinocyte cancers, comparing transplant recipients with and without these cancers, matched for gender, age, type of organ transplanted, post-transplantation period and immunosuppressive therapy. As indicated in the cover letter, a previous version was sent to JEADV and the authors have revised and supplemented the paper based on suggestions of the former submission. Each 194 transplant patients with and without NMSC were studied with respect to UV exposure and skin aging using a standardized questionnaire and validated visual score for measuring facial skin aging (SCINEXA). In addition genetic analysis of MR1C was performed in all cases. The risk to develop NMSC did not associate with the extent of skin aging, UV related skin damage or cumulative UV exposure. Only intensive UV exposure in the past (intentional recreational and whole body UV exposure) was associated with cancer. On the other hand, the risk for NMSC was shown to be significantly increased for patients with actinic keratosis, patients with green or blue eyes,as well as in patients with medium or high-risk MC1R variants. The authors conclude that in the absence of skin aging contributing to NMSC development, certain MC1R risk-types may identify OTR at risk for high tumor burden.
This comprehensive and carefully performed study shows that the risk of NMSC development in OTR cannot be assessed sufficiently by measuring skin aging, UV-related skin damage or cumulative UV exposure. Dermatologic follow-up of OTR with respect to development of skin cancer thus needs to include additional factors, as for instance, genetic changes like MC1R mutations. In this regard the authors present some intersting findings indicating specific MC1R variants or risk groups may be asscoiated with a higher risk for NMSC development. The manuscript fits well into the scope of the Special Issue which concerns with views and perspectives of cSCC.
I have just a few comments to make
Patients: a statement about ethical concerns should be given (approval of the ethics committee)
MC1R analysis: Although referred to supplementary files the definition of MC1R risk groups should be given in the main manuscript. The authors are asked to add a corresponding paragraph including significant references under 2.3 Genetic analyses (Methods, p.4).
PHP: the authors describe results of a pilot study using NGS to identify mtDNA mutations in a subset of 23 patients. Although this is an extensive experimental analysis, the results were presented in a very brief summary under 3.4. and in a supplementary table S12. The table is a somewhat difficult to understand. The number of mutations in all samples from UV exposed sites is 2 (is this the number of mutation in a certain number of nucleotides?) wheres in cases and controls it is 0 and 0.08 (I would assume the sum of both should be 2). Not unexpectedly, mutations were more frequently in UV exposed samples than in unexposed skin, but there is no significant difference between cases (cancer patients) and controls, which, however, may depend on the small number of tested specimens. I would suggest to delete these investigations from the paper, as they are still very preliminary and the manuscript already contains a lot of data.
Author Response
Alexandra Geusau, MD, Assoc. Professor
Department of Dermatology,
Medical University of Vienna
Waehringer Guertel 18-20
1090 Vienna, Austria
Email: Alexandra.geusau@meduniwien.ac.at
Tel. xx43-1-40400-77690
Fax xx43-1-40400-76990 25.1.2023
Cancers Editorial Office
MDPI, Poly Metropolitan
1201, Building 2, Courtyard 4, Guanyinan North Street, Tongzhou District,
101101 Beijing, China
Tel.: +86 10 6954 3724
E-mail: cancers@mdpi.com
Re: cancers-2144805 – response to reviewers
Cumulative UV exposure or a modified SCINEXA™-skin aging score do not play a substantial role in predicting the risk of developing keratinocyte cancers after solid organ transplantation – a case control study
by Liliane Borik-Heil,# Georg. Endler,# W. Parson, L. Schnaller, A. Zuckermann, K. Ünal, P. Jaksch,6 G. Böhmig, D. Cejka, K. Staufer, E. Hielle-Wittmann, S. Rasoul-Rockenschaub, P. Wolf, R. Sunder-Plassmann,+ A. Geusau,+*
# and + both authors contributed equally, *Corresponding author
Dear Editor,
Please find attached our revised manuscript ‘Cumulative UV exposure or a modified SCINEXA™-skin aging score do not play a substantial role in predicting the risk of developing keratinocyte cancers after solid organ transplantation – a case control study’ by Liliane Borik-Heil et al. for consideration to be published in ‘Cancers’ as ‘Original Article’ (special issue of Cancers "Views and Perspectives of Cutaneous Squamous Cell Carcinoma"), in a corrected version with all changes highlighted.
In addition, we have responded to all the reviewer's comments in the following text.
We are looking forward to your favourable reply.
Yours faithfully,
Alexandra Geusau
Comments to the reviewers’ remarks
First of all, we would like to thank all reviewers for their valuable time spent reading and evaluating our manuscript!
Reviewer: 1
Comments to the Author: I have just a few comments to make
- Patients: a statement about ethical concerns should be given (approval of the ethics committee)
- Are stated on page 17 in the main manuscript
Institutional Review Board Statement: This study was conducted according to the guidelines of the Declaration of Helsinki and approved by the Ethics Committee of the Medical University of Vienna (protocol number 1233/2016).
Informed Consent Statement: Informed consent was obtained from all subjects involved in the study.
- MC1R analysis: Although referred to supplementary files the definition of MC1R risk groups should be given in the main manuscript. The authors are asked to add a corresponding paragraph including significant references under 2.3 Genetic analyses (Methods, p.4).
We added the following paragraph as suggested under 2.3
Participants who did not carry an MC1R variant and individuals who carried variants that were unlikely to affect receptor function (based on published data or as predicted by bioinformatic mutation analysis tools) were considered as MC1R 0/0 risk group. Together with heterozygous carriers of MC1R r variants (0/r; reduced receptor function), these individuals were assigned to the MC1R low-risk group (0/0 plus 0/r). Medium genetic risk for MC1R dysfunction was defined by the presence of either heterozygous R (0/R; loss of function), compound heterozygous Rr, or homo- or compound heterozygous rr variants. Individuals assigned high MC1R risk carried homozygous or compound heterozygous RR variants.
- PHP: the authors describe results of a pilot study using NGS to identify mtDNA mutations in a subset of 23 patients. Although this is an extensive experimental analysis, the results were presented in a very brief summary under 3.4. and in a supplementary table S12. The table is a somewhat difficult to understand. The number of mutations in all samples from UV exposed sites is 2 (is this the number of mutation in a certain number of nucleotides?) wheres in cases and controls it is 0 and 0.08 (I would assume the sum of both should be 2). Not unexpectedly, mutations were more frequently in UV exposed samples than in unexposed skin, but there is no significant difference between cases (cancer patients) and controls, which, however, may depend on the small number of tested specimens. I would suggest to delete these investigations from the paper, as they are still very preliminary and the manuscript already contains a lot of data. Answer: This section has been abridged and mostly included in the supplement.

Reviewer 2 Report
Authors presented a case control study evaluating relationship of factors with NMSC. All necessary data are presented in supplementary materials and results are organized well in a main text. I suggest a minor revision before publication.
(1) Why did you use KC instead of NMSC in title?
(2) Importance or motivation of the research should be included in Abstract and Introduction.
(3) Typos should be checked such as mayor at line 195.
(4) I recommend to write each statistical analysis method in each table or graph caption.
Author Response
Alexandra Geusau, MD, Assoc. Professor
Department of Dermatology,
Medical University of Vienna
Waehringer Guertel 18-20
1090 Vienna, Austria
Email: Alexandra.geusau@meduniwien.ac.at
Tel. xx43-1-40400-77690
Fax xx43-1-40400-76990 25.1.2023
Cancers Editorial Office
MDPI, Poly Metropolitan
1201, Building 2, Courtyard 4, Guanyinan North Street, Tongzhou District,
101101 Beijing, China
Tel.: +86 10 6954 3724
E-mail: cancers@mdpi.com
Re: cancers-2144805 – response to reviewers
Cumulative UV exposure or a modified SCINEXA™-skin aging score do not play a substantial role in predicting the risk of developing keratinocyte cancers after solid organ transplantation – a case control study
by Liliane Borik-Heil,# Georg. Endler,# W. Parson, L. Schnaller, A. Zuckermann, K. Ünal, P. Jaksch,6 G. Böhmig, D. Cejka, K. Staufer, E. Hielle-Wittmann, S. Rasoul-Rockenschaub, P. Wolf, R. Sunder-Plassmann,+ A. Geusau,+*
# and + both authors contributed equally, *Corresponding author
Dear Editor,
Please find attached our revised manuscript ‘Cumulative UV exposure or a modified SCINEXA™-skin aging score do not play a substantial role in predicting the risk of developing keratinocyte cancers after solid organ transplantation – a case control study’ by Liliane Borik-Heil et al. for consideration to be published in ‘Cancers’ as ‘Original Article’ (special issue of Cancers "Views and Perspectives of Cutaneous Squamous Cell Carcinoma"), in a corrected version with all changes highlighted.
In addition, we have responded to all the reviewer's comments in the following text.
We are looking forward to your favourable reply.
Yours faithfully,
Alexandra Geusau
Comments to the reviewers’ remarks
First of all, we would like to thank all reviewers for their valuable time spent reading and evaluating our manuscript!
Reviewer 2
Comments to the Author: Authors presented a case control study evaluating relationship of factors with NMSC. All necessary data are presented in supplementary materials and results are organized well in a main text. I suggest a minor revision before publication.
- Why did you use KC instead of NMSC in title? Answer: The term KC is now frequently used; in particular when only cSCC and BCC are addressed, whereas the term NMSC also includes rare non melanoma skin tumor entities such as Merkel cell carcinoma. As mentioned for example in the recent review: Skin Cancer Metabolic Profile Assessed by Different Analytical Platforms. By Hagyousif YA, Sharaf BM, Zenati RA, El-Huneidi W, Bustanji Y, Abu-Gharbieh E, Alqudah MAY, Giddey AD, Abuhelwa AY, Alzoubi KH, Soares NC, Semreen MH. Int J Mol Sci. 2023 Jan 13;24(2):1604. doi: 10.3390/ijms24021604. doi: 10.3390/ijms24021604.
- Importance or motivation of the research should be included in Abstract or Introduction. Answer: We have added at the end of the introduction: In our study we aimed to investigate an association between environmental and genetic factors and skin cancer incidence in a carefully selected cohort of OTR. We assessed individual UV burden, quantified skin aging using a validated photoaging score, and determined the MC1R genotype of the participants.
- Typos should be checked such as mayor at line 195. done
- I recommend to write each statistical analysis method in each table or graph caption.
Done

Reviewer 3 Report
The authors have investigated the use of a validated photoaging score as a surrogate for risk of keratinocyte cancer development in organ transplant patients, a population that has a very large increased risk of these cancers. They find that other well validated markers of pigmentation are much better indicators of risk in this population, indeed the validated score was not a predictor. While this work is of use in itself, it would be better targeted to a dermatology journal.
Author Response
Alexandra Geusau, MD, Assoc. Professor
Department of Dermatology,
Medical University of Vienna
Waehringer Guertel 18-20
1090 Vienna, Austria
Email: Alexandra.geusau@meduniwien.ac.at
Tel. xx43-1-40400-77690
Fax xx43-1-40400-76990 25.1.2023
Cancers Editorial Office
MDPI, Poly Metropolitan
1201, Building 2, Courtyard 4, Guanyinan North Street, Tongzhou District,
101101 Beijing, China
Tel.: +86 10 6954 3724
E-mail: cancers@mdpi.com
Re: cancers-2144805 – response to reviewers
Cumulative UV exposure or a modified SCINEXA™-skin aging score do not play a substantial role in predicting the risk of developing keratinocyte cancers after solid organ transplantation – a case control study
by Liliane Borik-Heil,# Georg. Endler,# W. Parson, L. Schnaller, A. Zuckermann, K. Ünal, P. Jaksch,6 G. Böhmig, D. Cejka, K. Staufer, E. Hielle-Wittmann, S. Rasoul-Rockenschaub, P. Wolf, R. Sunder-Plassmann,+ A. Geusau,+*
# and + both authors contributed equally, *Corresponding author
Dear Editor,
Please find attached our revised manuscript ‘Cumulative UV exposure or a modified SCINEXA™-skin aging score do not play a substantial role in predicting the risk of developing keratinocyte cancers after solid organ transplantation – a case control study’ by Liliane Borik-Heil et al. for consideration to be published in ‘Cancers’ as ‘Original Article’ (special issue of Cancers "Views and Perspectives of Cutaneous Squamous Cell Carcinoma"), in a corrected version with all changes highlighted.
In addition, we have responded to all the reviewer's comments in the following text.
We are looking forward to your favourable reply.
Yours faithfully,
Alexandra Geusau
Comments to the reviewers’ remarks
First of all, we would like to thank all reviewers for their valuable time spent reading and evaluating our manuscript!
Reviewer 3
Comments to the Author: The authors have investigated the use of a validated photoaging score as a surrogate for risk of keratinocyte cancer development in organ transplant patients, a population that has a very large increased risk of these cancers. They find that other well validated markers of pigmentation are much better indicators of risk in this population, indeed the validated score was not a predictor. While this work is of use in itself, it would be better targeted to a dermatology journal. Answer: We consider the publication of this manuscript in a special issue on skin cancer to be appropriate.

Reviewer 4 Report
The authors present an interesting study which focusses on risk factors for patients after receiving organ transplants to develop non-melanoma skin cancer.
A huge amount of data is given, not all of them really necessary to mention. The authors should streamline the informations (e.g. signs of skin ageing increase with age).
The results are clearly worked out and are of relevance for patients receiving longterm immunosuppression.
However, the authors should speculate why the skin ageing score is not of relevance in predicting the individual skin cancer risk.
Author Response
Alexandra Geusau, MD, Assoc. Professor
Department of Dermatology,
Medical University of Vienna
Waehringer Guertel 18-20
1090 Vienna, Austria
Email: Alexandra.geusau@meduniwien.ac.at
Tel. xx43-1-40400-77690
Fax xx43-1-40400-76990 25.1.2023
Cancers Editorial Office
MDPI, Poly Metropolitan
1201, Building 2, Courtyard 4, Guanyinan North Street, Tongzhou District,
101101 Beijing, China
Tel.: +86 10 6954 3724
E-mail: cancers@mdpi.com
Re: cancers-2144805 – response to reviewers
Cumulative UV exposure or a modified SCINEXA™-skin aging score do not play a substantial role in predicting the risk of developing keratinocyte cancers after solid organ transplantation – a case control study
by Liliane Borik-Heil,# Georg. Endler,# W. Parson, L. Schnaller, A. Zuckermann, K. Ünal, P. Jaksch,6 G. Böhmig, D. Cejka, K. Staufer, E. Hielle-Wittmann, S. Rasoul-Rockenschaub, P. Wolf, R. Sunder-Plassmann,+ A. Geusau,+*
# and + both authors contributed equally, *Corresponding author
Dear Editor,
Please find attached our revised manuscript ‘Cumulative UV exposure or a modified SCINEXA™-skin aging score do not play a substantial role in predicting the risk of developing keratinocyte cancers after solid organ transplantation – a case control study’ by Liliane Borik-Heil et al. for consideration to be published in ‘Cancers’ as ‘Original Article’ (special issue of Cancers "Views and Perspectives of Cutaneous Squamous Cell Carcinoma"), in a corrected version with all changes highlighted.
In addition, we have responded to all the reviewer's comments in the following text.
We are looking forward to your favourable reply.
Yours faithfully,
Alexandra Geusau
Comments to the reviewers’ remarks
First of all, we would like to thank all reviewers for their valuable time spent reading and evaluating our manuscript!
Reviewer 4
Comments to the Author:
- The authors present an interesting study which focusses on risk factors for patients after receiving organ transplants to develop non-melanoma skin cancer.
A huge amount of data is given, not all of them really necessary to mention. The authors should streamline the informations (e.g. signs of skin ageing increase with age).
Answer:
See Table 4 (‘Factors associated with skin aging, NMSC development and NMSC number’):
We have added to the conclusion:
To summarize, in OTR, skin aging increases with age, smoking history and high UVR exposure. Men have higher skin aging-scores. Risk factors for developing NMSC include lighter phenotype, holiday-related high UVR, specific MC1R variants and the presence of AK, but not skin aging per se. The number of NMSC increases with longer time after TX period, shorter interval between TX and first NMSC, older age, higher skin aging scores, presence of AK, specific MC1R variants, and severe sunburns as young adults.
- The results are clearly worked out and are of relevance for patients receiving long-term immunosuppression. However, the authors should speculate why the skin ageing score is not of relevance in predicting the individual skin cancer risk.
Answer:
As requested by the reviewer, we have added the following sentence to the discussion: In addition to exogenous factors, interindividual differences in skin aging can also be attributed to genetic factors that influence collagen degradation and elastin deposition (). The intrinsic skin aging score may not adequately reflect these factors, which could explain the lack of association between skin aging and KC development. We have also added the respective citations.

Round 2
Reviewer 3 Report
None